# Graph-Based Data Fusion Applied to: Change Detection and Biomass Estimation in Rice Crops

**David Alejandro Jimenez-Sierra [1,\*] , Hernán Darío Benítez-Restrepo [1] ,**
**Hernán Darío Vargas-Cardona [1] and Jocelyn Chanussot [2]**

[1] Departamento de Electrónica y Ciencias de la Computación, Pontificia Universidad Javeriana Seccional Cali, Cali 760031, Colombia; hbenitez@javerianacali.edu.co (H.D.B.-R.); hernan.vargas@javerianacali.edu.co (H.D.V.-C.)

[2] Grenoble Images Parole Signals Automatique Laboratory (GIPSA-Lab), Grenoble Institute of Technology, 38031 Grenoble, France; jocelyn.chanussot@gipsa-lab.grenoble-inp.fr

\* Correspondence: davidjimenez@javerianacali.edu.co

**Abstract:** The complementary nature of different modalities and multiple bands used in remote sensing data is helpful for tasks such as change detection and the prediction of agricultural variables. Nonetheless, correctly processing a multi-modal dataset is not a simple task, owing to the presence of different data resolutions and formats. In the past few years, graph-based methods have proven to be a useful tool in capturing inherent data similarity, in spite of different data formats, and preserving relevant topological and geometric information. In this paper, we propose a graph-based data fusion algorithm for remotely sensed images applied to (i) data-driven semi-unsupervised change detection and (ii) biomass estimation in rice crops. In order to detect the change, we evaluated the performance of four competing algorithms on fourteen datasets. To estimate biomass in rice crops, we compared our proposal in terms of root mean squared error (RMSE) concerning a recent approach based on vegetation indices as features. The results confirm that the proposed graph-based data fusion algorithm outperforms state-of-the-art methods for change detection and biomass estimation in rice crops.

**Keywords:** biomass estimation; change detection; data fusion; graph based; multi-modal; multi-temporal; multi-spectral; remote sensing

## 1. Introduction

Recent advances in sensor technology have led to the increased availability of hyper-spectral, multi-spectral (MS), and synthetic aperture radar (SAR) images (at very high spatial and spectral resolutions), which describe an object or phenomenon. Each sensor captures different information that explains physical features. For example, a SAR sensor captures information about the physical characteristics of a surface (such as roughness, geometric structure, and orientation), and an MS sensor captures reflectances at different wavelengths from objects. Therefore, it is generally desirable to use more sensors rather than fewer [1]. For hyper-spectral and multi-spectral images, the fusion approaches can be categorized into component substitution, multi-resolution analysis, unmixing, and Bayesian-based methods. We encourage the reader to refer to [2] for a comprehensive review. Even though data fusion contributes to better performance in classification and detection in remote sensing, it is a complex task. For example, the different resolutions, units, dimensions, and formats are challenges imposed by raw data [3]. Thus, the extraction of features helps to cope with those challenges. Additionally, in recent years, graph-based fusion algorithms have been able

to tackle the variability of data formats and provide a flexible way of representing the relationship between data entities [4]. Unsurprisingly, graph-based approaches have also been applied to the task of data fusion [5–8]. For instance, the authors in [9] proposed a graph-based data fusion (GDF) method to couple data and dimension reduction for the classification of multi-sensor imagery. GDF [9] combines multiple feature sources through a fused graph. However, this approach requires big storage capacity and considerable computational load to process large training datasets. As an illustration, to process an image of size $3000 \times 2000$, approximately 67 GB of RAM are needed. Additionally, the final fused graph contains the same weights (binary matrix) for all connections among nodes, which is not always realistic. Moreover, the fusion rule utilized for the graphs in the aforementioned study (called the element-wise product) lacks generalization. A generalized version of GDF [10] tries to solve the interaction among nodes by using a new metric space to weigh the connected nodes of the graph with a Gaussian kernel. However, the considerable computational load is still a problem for GDF implementation. As a reasonable solution to this issue, the authors in [11] proposed an approach using a sliding window for fusing local graphs across their intersection. Nevertheless, this local approach treats all the connected nodes as equal (binary matrix), which is not always true. Furthermore, the fusion rule only considers the shared connections in order to preserve coherence in the fusion. However, this rule does not capture relevant features that could be explained by the relationship between the nodes, which are not strictly the same shared connections. In [1], the authors proposed an approximated global graph with non-equal weights (non-binary matrix) by using the Nyström extension to generate a graph to fuse RGB and LiDAR images. The authors fused a stacked version of the datasets. Then, they computed the graph and classified urban areas by applying *k*-means on the eigenvectors of the graph related to the fused data. In this paper, we extend this idea by introducing a mutual information criterion that selects the most representative eigenvector that captures the variability of the fused data. To illustrate the generalization of the proposed model, we apply it in two different tasks: change detection and biomass estimation of rice crops.

## 1.1. Change Detection

Change detection (CD) refers to the task of analyzing two or more images acquired over the same area at different times (multi-temporal images), with the aim of detecting zones in which the land-cover type changed between the acquisitions [12]. CD makes it possible to quantify the magnitude of natural disasters (such as flooding) and changes generated by human activity. This analysis provides fundamental data for environmental protection, sustainable development, and the maintenance of ecological balance [13]. CD deploys inputs known as multi-spectral (MS) images that contain information from both the spatial and spectral domains (such as sensors in the Landsat series of satellites). Providing two or more co-registered images, pixel-based approaches carry out change detection using probabilistic thresholding and machine learning methods [14,15]. Although thresholding methods are efficient and useful, they are also sensitive to MS image noise and require a high degree of accuracy in the estimation of the probabilistic distribution of the difference image. These issues make thresholding methods prone to artifacts in the final change map [16–20]. Machine learning methodologies are divided into two categories: classification and clustering. Classification methods require a multitemporal reference, which is difficult to extract from raw data, so these methods are not a practical solution [21]. Clustering techniques [22–26] are affected by parameter initialization, which may generate local minima in the learning stage. In addition, the intrinsic brightness distortion in MS images yields inaccurate change maps [15]. Furthermore, deep learning (DL) approaches also are used in change detection [27]. These methods are based on autoencoders (AEs), deep belief networks (DBNs), convolutional neuronal networks (CNNs) [28], recurrent neural networks (RNNs), pulse coupled neural networks (PCNNs), and generative adversarial networks (GANs) [27,29]. Nevertheless, DL approaches present issues such as the over-fitting of data when the training dataset is small and the optimization of hyper-parameters [27,29]. Moreover, multi-modality (inputs from different sensors) is an important challenge for CD. For instance, data representation by

heterogeneous physical units [30,31] has been addressed by processing techniques (such as domain adaptation, data transformation, transfer learning, and image-to-image translation [2,26,31–33]) in such a way that datasets lying in different domains are brought into one single domain for comparison.

In order to reduce the effects of small inter-class variability and artifacts presented in MS images, we propose a graph-based data fusion methodology that works with both heterogeneous and homogeneous data. We validated our approach using fourteen real cases.

### 1.2. Biomass Estimation

The measurement of biomass in rice crops relies on destructive sampling or satellite image analysis. Destructive sampling involves much manual work to gather plant samples. Subsequently, it is necessary to measure the accumulated dry weight determined by leaves, stems, panicles, and all the aboveground components of rice canopies, per unit of a given area in the field [34]. The remote sensing approach, on the other hand, uses the information sourced from satellites, which provide crop-scale images with limited resolution, to perform non-invasive image-based crop data estimation. In addition, unmanned aerial vehicles (UAVs) offer a number of benefits; firstly high-resolution information, secondly relevant relationships between vegetation indices, photosynthetic activity, and canopy structural properties, and thirdly, reliable aboveground biomass estimation (AGBE) [35–38]. In the last few years, the low cost and flexibility of UAVs have created new opportunities in precision agriculture and phenotyping. They have made it possible to calculate vegetation indices (VI) from multi-spectral and thermal imagery captured from the crop. For instance, the normalized difference vegetation index (NDVI) fuses reflectances from the red band (R) and near-infrared (NIR) and is one of the most popular VIs used by farmers to quantify crop vegetation density. Although NDVI is accurate in the early stages of a crop [35], it saturates as the biomass grows. This issue produces inaccurate measurements during late growth crop stages [39]. Nonetheless, a combination of several VIs can improve the assessment of the impact that each stage of plant growth has on crop yield [40,41].

Given the advantages of UAVs with respect to alternative methods for gathering data (such as manual collection or satellite image analysis [37,38] in agriculture applications), they have become an excellent alternative for crop monitoring. Several methods have been proposed for AGBE [38,42–44], which have at their core the training of machine learning methods based on features extracted from vegetation indices (VIs). A recent approach presented in [38] pre-processes MS images to extract the pixels corresponding to the rice crop. It then uses VIs to train three separate linear regression models for each growth stage (vegetative, reproductive, and ripening). To build a unique model that captures the variability of biomass for all the growth stages, we propose the use of eigenvectors as features extracted from a fused graph.

This paper is structured as follows. The next section details the proposed graph-based fusion method and each step involved in the applications: change detection and estimation of biomass in rice crops. Section 3 presents the experimental results that verify the effectiveness of the proposed approach on fourteen different real remote sensing datasets for detecting the changes and one real dataset to estimate biomass in rice crops. In Section 4, we set out conclusions.

## 2. Materials and Methods

Since graphs explain the structural relationships between nodes (such as image pixels) and also capture local information related to data (such as radiometric similarities), the proposed graph-based data fusion approach aims to:

- Construct an approximate local graph related to remotely sensed images (such as an MS image captured by Landsat/UAV) by using the Nyström extension.
- Perform data fusion over the local graphs by minimizing the similarity between the connections of the graphs to capture relevant information embedded in the case studies.

- Extract different relationships given in the fused data by computing the eigenvectors/eigenvalues of the fused graphs.
- Evaluate the performance of the proposed graph-based data fusion in the applications of change detection and biomass estimation.

*2.1. Graph-Based Data Fusion*

MS images contain pixels that reside on a regularly sampled 2D grid. Thus, it is possible to interpret them as a signal on a graph with edges that connect each pixel in each band to its neighborhood of pixels. A graph is a nonlinear structural representation of data, defined by $G = (V, E)$, where $G$ is the graph, $V$ is a set of nodes, and $E$ refers to the arcs or edges that explain the directed or undirected relationship between nodes. The edges have an associated weight of $w_{i,j}$, which quantifies how strong the relationship is between the nodes. The common measure used for each weight is a Gaussian kernel [45]:

$$w_{i,j} = \exp\left(-\frac{d(V_i, V_j)^2}{\sigma^2}\right),$$ (1)

where $d(V_i, V_j)$ is the distance between two nodes and $\sigma$ is the standard deviation of all $d(V_i, V_j)$. A common application for graphs is the embedding of $G$ based on the Laplacian (**L**) matrix into a space $\mathbb{R}^m$. That keeps the graph nodes as close as they were in the input space. In short, the embedding of a graph is given by the eigen problem $\mathbf{L}\mathbf{y} = \lambda \mathbf{D}\mathbf{y}$ [46], where $\mathbf{L} = \mathbf{D} - \mathbf{W}$, **W** is known as the adjacency matrix, or weights of the graph (each component is given by Equation (1)), and **D** is a diagonal matrix, its components being the degree of the node ($di = \sum_j w_{i,j}$).

As there is such a high number of pixels in an MS image, the computational cost of calculating the full matrix $\mathbf{W} \in \mathbb{R}^{N \times N}$ is extremely high (an image with a resolution of $1280 \times 960$ is equivalent to $N = 1{,}228{,}800$). To solve this problem, we apply the Nyström extension [47] to find an approximation of **W** in significantly reduced time. To select points uniformly distributed across the image, $n_s$ samples are selected by spatial grid sampling and re-indexing the matrix **W** as:

$$\mathbf{W} = \kappa_G\left(\begin{bmatrix} \mathbf{d_{AA}} & \mathbf{d_{AB}} \\ \mathbf{d_{AB}}^\top & \mathbf{C} \end{bmatrix}\right),$$ (2)

where $\kappa_G$ is a Gaussian kernel, $\mathbf{d_{AA}} \in \mathbb{R}^{n_s \times n_s}$ represents the graph distances within the $n_s$ sample nodes, $\mathbf{d_{AB}} \in \mathbb{R}^{n_s \times (N - n_s)}$ are the distances between the $n_s$ sample nodes and the remaining $N - n_s$ nodes, and $\mathbf{C} \in \mathbb{R}^{(N - n_s) \times (N - n_s)}$ are the distances within the unsampled nodes. This method approximates **C** by choosing $n_s$ samples uniformly distributed across the image from the dataset of size $N$ ($n_s \ll N$), hence:

$$\mathbf{W} \approx \widehat{\mathbf{W}} = \kappa_G\left(\begin{bmatrix} \mathbf{d_{AA}} \\ \mathbf{d_{AB}} \end{bmatrix}\right).$$ (3)

Thus, the eigenvectors of the matrix $\widehat{\mathbf{W}}$ can be spanned by the eigenvalues and eigenvectors of $\kappa_G(\mathbf{d_{AA}})$. Solving the diagonalization of $\kappa_G(\mathbf{d_{AA}})$ (eigenvalues $\lambda$ and eigenvectors **U**: $\kappa_G(\mathbf{d_{AA}}) = \mathbf{U}^\top \mathbf{\Lambda} \mathbf{U}$), the eigenvectors of $\widehat{\mathbf{W}}$ can be spanned by:

$$\hat{\mathbf{U}} = \begin{bmatrix} \mathbf{U} \\ \kappa_G(\mathbf{d_{AB}})^\top \mathbf{U} \mathbf{\Lambda}^{-1} \end{bmatrix}.$$ (4)

Since the approximated eigenvectors $\hat{\mathbf{U}}$ are not orthogonal, as explained in [47], to obtain orthogonal eigenvectors, we use $\mathbf{S} = \kappa_G(\mathbf{d_{AA}}) + \kappa_G(\mathbf{d_{AA}})^{-\frac{1}{2}} \kappa_G(\mathbf{d_{AB}}) \kappa_G(\mathbf{d_{AB}})^\top \kappa_G(\mathbf{d_{AA}})^{-\frac{1}{2}}$. Then, by diagonalization of **S** ($\mathbf{S} = \mathbf{U_s} \mathbf{\Lambda_s} \mathbf{U_s}$), the final approximated eigenvectors of **W** are given by:

$$\hat{\mathbf{U}} = \begin{bmatrix} \kappa_G(\mathbf{d_{AA}}) \\ \kappa_G(\mathbf{d_{AB}})^\top \kappa_G(\mathbf{d_{AA}})^{-\frac{1}{2}} \end{bmatrix} \mathbf{U_s} \mathbf{\Lambda_s}^{-\frac{1}{2}}.$$ (5)

Fusion Stage

In this section, we present the fusion stage to integrate the multi-temporal data. We model each pixel as a node in the graph and assume that pre-event and post-event images contain the same number of elements and that they are co-registered. Figure 1 presents a diagram of the method explained in Algorithm 1, which processes each instance of band $b$ and time $k$ ($\mathbf{X}^{b,k}$) and the number of samples $n_s$ as inputs.

---

**Algorithm 1:** GBF for temporal data.

---

**Input:** Temporal images from band $b$ or set of bands $\mathbf{X}^{b,k} \in \mathbb{R}^{m \times n}$, number of samples $n_s$
**Output:** Fused graph $\mathbf{W_F} \in \mathbb{R}^{(n_s+c) \times n_s}$
**Initialize:** $k = 1$, $N = m \times n$
**while** $k \leq 2$ **do**

(1) Scale the data by $\mathbf{X}^{b,k} = \frac{\mathbf{X}^{b,k}}{max(\mathbf{X}^{b,k})}$

(2) Take $n_s$ samples uniformly distributed across $\mathbf{X}^{b,k}$ by spatial grid sampling.
　　$X_{AA}^{b,k} = \text{sampler}(\mathbf{X}^{b,k}, n_s), \in \mathbb{R}^{n_s}$

(3) Find the complement $\overline{X}^{b,k} \in \mathbb{R}^c$ of $X_{AA}^{b,k}$ in $\mathbf{X}^{b,k}$.

(4) For each set $X_{AA}^{b,k}$ and $\overline{X}^{b,k}$, perform the pairwise distance between samples-samples $(\mathbf{d_{AA}}^{b,k} \in \mathbb{R}^{n_s \times n_s})$ and samples-complement $(\mathbf{d_{AB}}^{b,k} \in \mathbb{R}^{c \times n_s})$.

$$\mathbf{d_{AA}}^{b,k} = \left\{ \left\| x_{AA_i}^{b,k} - x_{AA_j}^{b,k} \right\|_2 \right\}_{i \quad j}^{n_s n_s}, \forall i \neq j$$

$$\mathbf{d_{AB}}^{b,k} = \left\{ \left\| \overline{x}_i^{b,k} - x_{AA_j}^{b,k} \right\|_2^3 \right\}_{i \quad j}^{c \quad n_s}, \forall i \neq j$$

(5) Apply the normalized graph Laplacian ($\hat{\mathbf{D}}^{-\frac{1}{2}} \widehat{\mathbf{W}} \hat{\mathbf{D}}^{-\frac{1}{2}}$) by using the code in [47].

(6) Apply a Gaussian kernel ($\kappa_G(.)$) with $\sigma = mean(\mathbf{d_{AB}})$ on the normalized distances, and build the approximated normalized Laplacian matrix based on the Nyström approximation.

$$\widehat{\mathbf{W}}_N^{b,k} = \left[ \kappa_G(\mathbf{d_{AA}}^{b,k}); \kappa_G(\mathbf{d_{AB}}^{b,k}) \right]^\top$$

(7) $k = k + 1$

**end**

$\mathbf{W_F} = min(\widehat{w}_{N_{ij}}^{b,k})$, with $i = 1, .., c; j = 1, .., n_s$.

---

The output of Algorithm 1 for one instance of time of a selected band or bands $X^{b,k}$ corresponds to the approximate normalized adjacency matrix ($\widehat{\mathbf{W}}_N^{b,k}$) [47]. Consequently, the fusion step consists of capturing the unique information given by each graph ($\widehat{\mathbf{W}}_N^{b,k}$) into one fused graph ($\mathbf{W_F}$). In order to achieve this fusion, we maximize the distance (or minimize the similarity) among the pixels:

$$\mathbf{W_F} = min(\widehat{w}_{N_{ij}}^{b,k}), \text{with } k = [1, 2],$$

where $w_{i,j}$ represents the weight of the node for each instance of time ($i = 1, 2, \ldots, c; j = 1, 2, \ldots, n_s$). This learning approach is data driven (uses a few uniformly distributed $n_s$ samples across the image). It is restarted for each dataset. The graph $\mathbf{W_F}$ represents the relationship in terms of dissimilarity between the pre-event and post-event images.

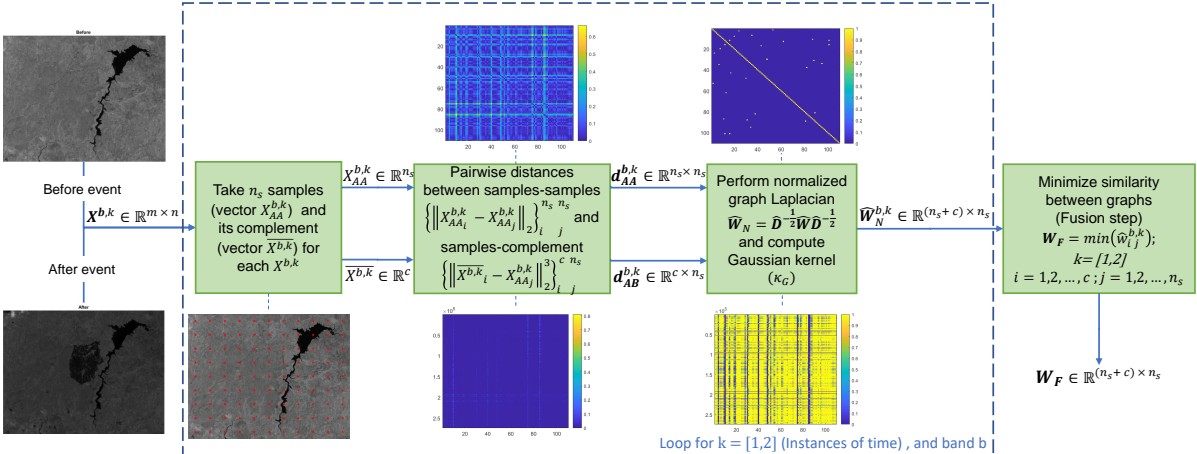

**Figure 1.** Graph-based fusion, where $k$ is the time of Events 1 (pre) and 2 (post), $b$ refers to the band, $X^{b,k}$ is an image that represents an event, $X^{b,k}_{AA}$ represents the samples from $X^{b,k}$, $\overline{X}^{b,k}$ is the complement, $\mathbf{d_{AA}}^{b,k}$ is the pairwise distance between the samples in $X^{b,k}_{AA}$, $\mathbf{d_{AB}}^{b,k}$ is the pairwise distance between $X^{b,k}_{AA}$ and $\overline{X}^{b,k}$, $\widehat{\mathbf{D}} = Diag(d_1, d_2, \ldots, d_{n_s})$ with $d_i = \sum_j^{n_s} \hat{w}^{b,k}_{ij}$ is the approximated degree matrix, and $\widehat{\mathbf{W}}^{b,k}_N$ is the normalized Laplacian calculated by using the Nyström approximation.

## 2.2. Change Detection Scheme Based on the Multi-Temporal Graph (GFB-CD)

The change detection scheme presented in Figure 2 uses the approximated eigenvectors and eigenvalues found by Nyström's extension from $\mathbf{W_F}$, as features to represent the change between the pre-event and post-event images. The number of eigenvectors is equal to the number of samples ($n_s$) taken from an instance of time $k$. To build the change map, we select the scaled eigenvector ($\mathbf{I}_{u_i}$) that maximizes the mutual information (MI) [48] of this eigenvector with a binarized prior signal. The prior signal comes from the normalized differences between pre-event and post-event images.

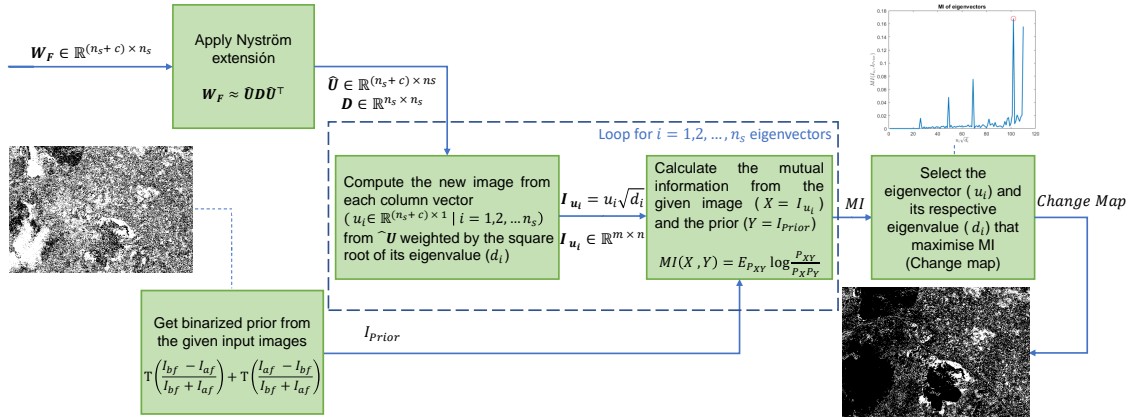

**Figure 2.** Change detection, where $\widehat{\mathbf{W}}_{\mathbf{F}}$ is the fused graph, $\widehat{\mathbf{U}}$ is the approximated eigenvectors, $\mathbf{D}$ is the eigenvalues, and $T$ in the prior stands for a binarization operator.

## 2.3. Graph-Based Fusion Regression for Estimating Biomass in Rice Crops

In terms of image processing, the analysis of images related to crops implies important challenges. Weather conditions can affect the quality of the data (sunny or cloudy). Another important factor is the appearance (architectural morphology) of the plant as it grows. The development of tillers occurs in the vegetative stage; the number of leaves increases, as well as the height of the plant. In this stage, the color green is predominant. In the reproductive stage, panicle formation starts, and thus, yellow features appear in the images. In the final ripening stage, the flowering, the grain filling, and the maturation of the plant occur, while the leaves begin to senesce. In this stage, the color yellow is

predominant, and the plot can barely be distinguished from panicles, while grains and senescent leaves predominate. In conclusion, it is possible to observe (see Figure 5) that during a plant's growth, it becomes more difficult to separate plots and distinguish between plants and background, using RGB images. Therefore, general assumptions about the color, the size of the plant, and the color of the soil will not always be correct [38]. Considering these limitations, we believe that the graph-based fusion of MS bands provides a flexible way of representing useful combinations of surface reflectance, to produce features at two or more wavelengths that predict biomass in rice crops at different stages of growth. We developed our method inspired by the work in [38], in which the authors estimated rice biomass as a function of one of the growth stages (vegetative, reproductive, and ripening). They proposed three models of linear regressions, one for each stage of the crop. Those models have inputs that are features extracted from VIs. A comprehensive survey of the specialized literature was carried out, in order to identify which vegetation indices are suitable for estimating rice biomass as a function of the growth stage of the crop [40–42,49]. The results of this survey are summarized in Table 1:

**Table 1.** VIs for biomass estimation.

| Name | Equation |
|---|---|
| Ratio Vegetation Index (**RVI**) [40] | $\frac{NIR}{RED}$ |
| Difference Vegetation Index (**DVI**) [50] | $NIR - RED$ |
| Normalized DVI (**NDVI**) [40] | $\frac{NIR - RED}{NIR + RED}$ |
| Green NDVI (**GNDVI**) [41] | $\frac{NIR - GREEN}{NIR + GREEN}$ |
| Corrected Transformed Vegetation Index (**CTVI**) [50] | $\frac{NDVI + 0.5}{|NDVI + 0.5|} \sqrt{|NDVI + 0.5|}$ |
| Soil-Adjusted Vegetation Index (**SAVI**) [41] | $(1 + L)\left(\frac{NIR - RED}{NIR + RED + L}\right)$, with $L = 0.5$ |
| Modified SAVI (**MSAVI**) [49] | $\frac{1}{2}(2NIR) + 1 - \sqrt{(2NIR + 1)^2 - 8(NIR - RED)}$ |

The following is a brief explanation of the procedure used by the authors in [38]: (i) segment the area covered by the crop from the soil by using $k$-means clustering ($K = 2$); (ii) extract VIs (features) from the crop pixels; (iii) train a linear regression model for each stage of the crop.

Firstly, the bands ($\mathbf{X}^{b,k}$) that are to be fused are red (R), green (G), and near-infrared (NIR) ($b = [1, 2, 3]$). Secondly, there are $n_s$ eigenvectors for each fused graph ($\mathbf{W_F}$). For each graph, we took the eigenvector with the associated highest eigenvalue, as it provides the strongest contribution to the Laplacian reconstruction. Thirdly, we stacked all these features as row vectors from each image into a matrix $\mathbf{F} \in \mathbb{R}^{q \times (n_s + c)}$. Fourthly, since there are $q = 489$ images with a size of $1280 \times 960$, the dimensionality of the features is high ($\approx$1.2 million dimensions). Consequently, we reduced the number of features to $z$ dimensions by applying two well-known techniques: principal component analysis (PCA) [51] and $t$ distributed stochastic neighbor embedding ($t$-SNE) [52]. Lastly, we trained a support vector machine (SVM) regressor [53] with a Gaussian kernel to predict the biomass over all growth.

Figure 3 illustrates our proposed method for estimating biomass in rice crops based on Algorithm 1 (setting $k = 1$ and $b = [1, 2, 3]$) and the graph-based fusion methodology shown in Figure 1.

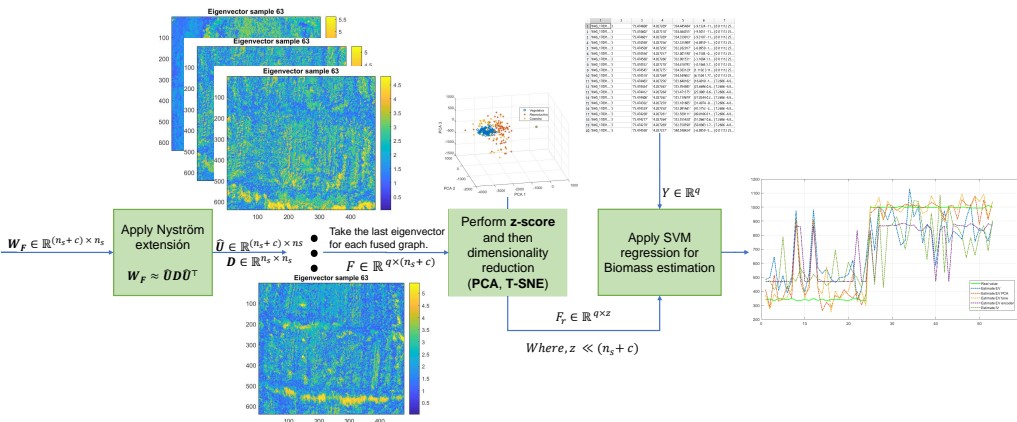

**Figure 3.** The proposed methodology based on graph fusion for estimating biomass in rice crops, from $q$ images.

The procedure used to train the models is given in Algorithm 2 below.

---

**Algorithm 2:** SVM-regression models, trained on features extracted from the fused graph.

---

**Input:** Images of rice crops to train $\mathbf{I_i} \in \mathbb{R}^{m \times n}$, number of images $q$,
biomass related to the image $Y \in \mathbb{R}^q$
**Output:** Regression models: $\mathbf{M_{TSNE}}$ and $\mathbf{M_{PCA}}$
**Initialize:** $N = m \times n$, $i = 1$, $z = 16$

**while** $i \leq q$ **do**

    (1) Execute Algorithm 1 with $X^{b,k} = I_i$, with $b = [1,2,3]$, $k = 1$, and $n_s = 100$.

    (Output $\mathbf{W_F}^i$)

    (2) Take the eigenvector with the highest associated eigenvalue of the fused graph:

    $v = \hat{\mathbf{U}}^i(:, n_s) \in \mathbb{R}^N$.

    (3) Stack the vector as features for the regression $F(:, i) = v^\top \in \mathbb{R}^{1 \times N}$

    $i = i + 1$

**end**

(4) Apply the z-score to the features.

(5) Perform dimensionality reduction of features ($\mathbf{F} \in \mathbb{R}^{q \times N}$):

$\mathbf{F_{TSNE}} = tsne(\mathbf{F}, z) \in \mathbb{R}^{q \times z}$, $\mathbf{F_{PCA}} = pca(F, z) \in \mathbb{R}^{q \times z}$

(6) Train the models with SVM-regression:

$\mathbf{M_{TSNE}} = fitrsvm(\mathbf{F_{TSNE}}, Y)$, $\mathbf{M_{PCA}} = fitrsvm(\mathbf{F_{PCA}}, Y)$

---

The outputs of Algorithm 2 are two regression models that predict the biomass related to an image of rice crops. These models use the reduced dimensions as inputs (such as PCA or *t*-SNE) of the eigenvectors from the fused graph with respect to the red, green, and near-infrared bands. The reason for applying the z-score is to avoid the high variability of features given for the entire growth stage of the rice crops and to decrease unstable biomass estimations.

*2.4. Datasets' Description*

Here, we describe the datasets used to measure the performance of the proposed graph-based data fusion method. For the change detection application, we considered fourteen real scenes captured by MS and SAR sensors (as shown in Table 2 and Figure 4), which include events such as: earthquakes, floods, wildfires, melted ice, farming, and building. In addition, for the biomass estimation task, we used 560 UAV images with their corresponding value of biomass measured by the destructive method.

**Table 2.** Databases used to evaluate the performance of the proposed method.

| Place | Event | Pre-Date | Post-Date | Lat | Lon | Size | Band | Sensor |
|---|---|---|---|---|---|---|---|---|
| Sardinia Island | Flood | 3 September 1995 | 3 July 1996 | 39.68, 39.55 | 9.10, 9.30 | 479 × 573 | NIR | Landsat-5 TM |
| Omodeo lake | Fire | 25 June 2013 | 10 August 2013 | 40.17, 39.97 | 8.66, 9.00 | 742 × 965 | RED | Landsat-5 TM |
| Alaska | Melt Ice | 24 June 1985 | 13 June 2005 | 70.761, 70.641 | −153.074, −152.553 | 443 × 642 | NIR | Landsat-5 TM |
| Brasil, Madeirinha | Farming building | 15 July 2000 | 16 July 2006 | −9.335, −9.433 | −61.942, −61.798 | 364 × 527 | RED | Landsat-5 TM |
| Colombia, Katios National Park | Fire | 10 March 2019 | 27 April 2019 | 7.943, 7.832 | −77.23, −77.063 | 879 × 1319 | SAR | Sentinel 1 A |
| Colombia, Atlantico | Flood (dam) | 28 April 2010 | 16 March 2011 | 10.439, 10.288 | −75.14, −74.921 | 729 × 1056 | SAR | ALOS/PALSAR |
| San Francisco | Flood | 10 August 2003 | 16 May 2004 | 38.11, 38.00 | 121.41, 122.46 | 275 × 400 | SAR | ERS-2 SAR |
| China, WenChuan | Earthquake | 3 March 2008 | 16 June 2008 | 31.049, 31.011 | 103.525, 103.581 | 301 × 442 | SAR | ESA/ASAR |
| France, Toulouse | Building | 10 February 2009 | 15 July 2013 | 43.5835, 43.5702 | 1.4318, 1.4817 | 2604 × 4404 | SAR/NIR | TerraSAR-X        Pleiades |
| Canada, Prince George | Fire | 6 July 2017 | 22 August 2017 | 51.48, 50.80 | −121.626, −120.863 | 2479 × 1905 | NIR | Landsat-8 |
| California | Flood | 11 January 2017 | 26 February 2017 | 39.346, 39.348 | −121.161, −121.924 | 3500 × 2000 | NIR/SAR | Landsat-8 Sentinel 1 A |
| U.K., Gloucester-1 | Flood | 5 September 1999 | 17 November 2000 | 52.126, 52.134 | −2.113, −2.280 | 4220 × 2320 | NIR | SPOT |
| Bastrop | Fire | 8 September 2011 | 22 October 2011 | 30.1316, 30.1321 | −97.2898, −97.3182 | 1534 × 808 | NIR/NIR | Landsat-5 TM        EO-1 ALI |
| U.K., Gloucester-2, UK | Flood | 14 June 2006 | 25 July 2007 | 51.8552, 51.8512 | −2.2174, −2.1910 | 4220 × 2320 | NIR/SAR | Quickbird 02 TerraSAR-X |

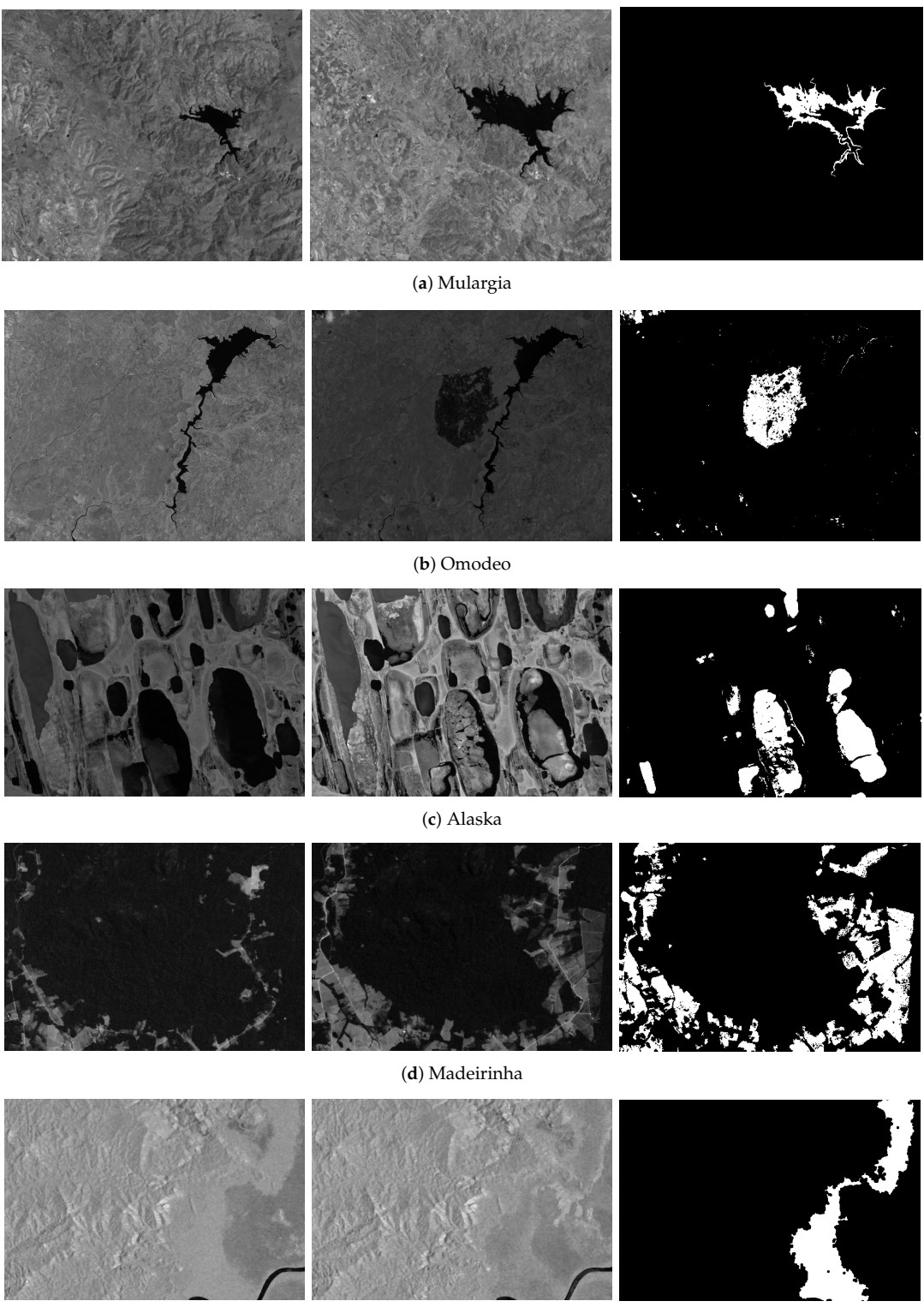

(**a**) Mulargia

(**b**) Omodeo

(**c**) Alaska

(**d**) Madeirinha

(**e**) Katios National Park

**Figure 4.** *Cont.*

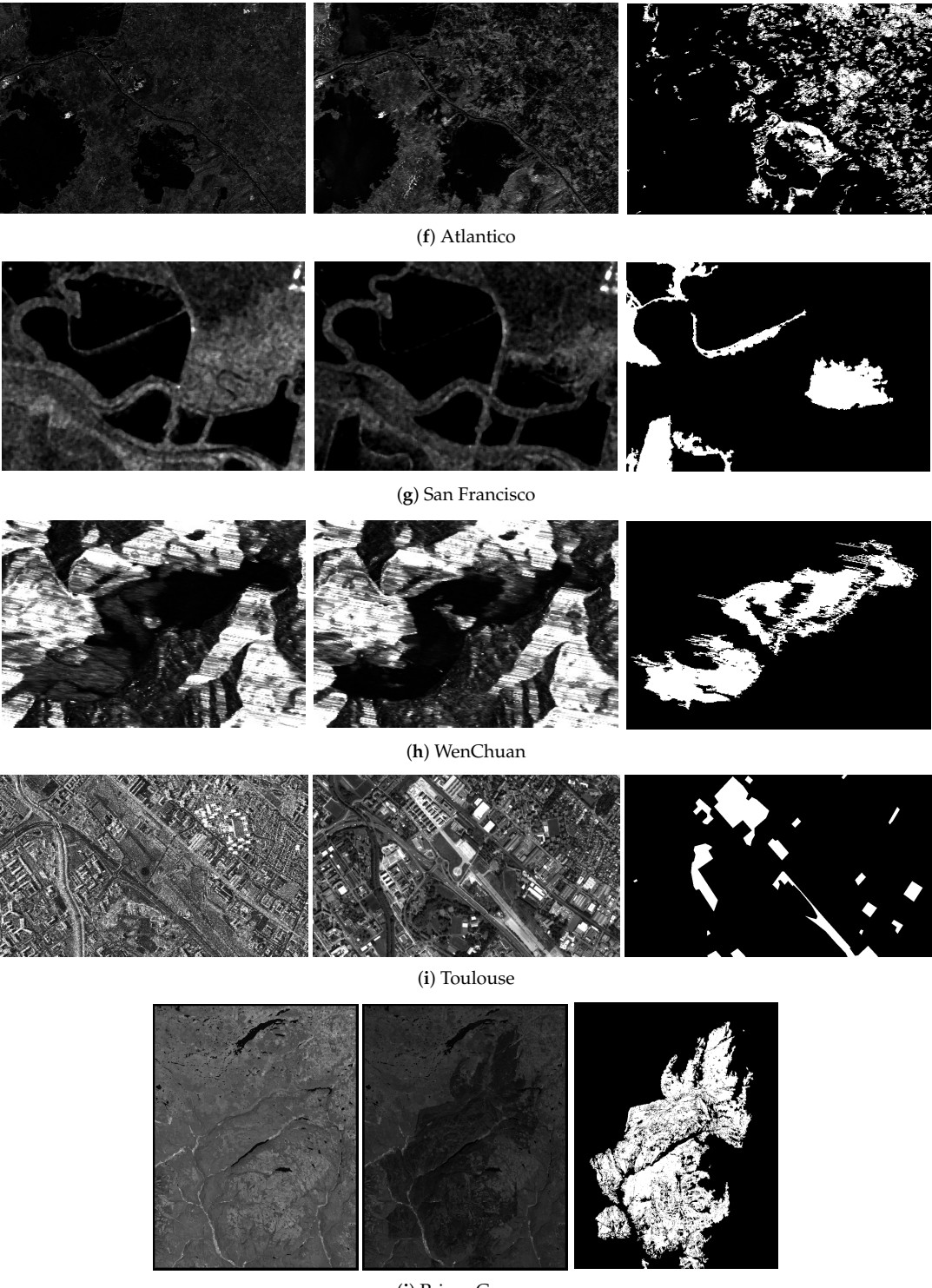

(**f**) Atlantico

(**g**) San Francisco

(**h**) WenChuan

(**i**) Toulouse

(**j**) Prince George

**Figure 4.** *Cont.*

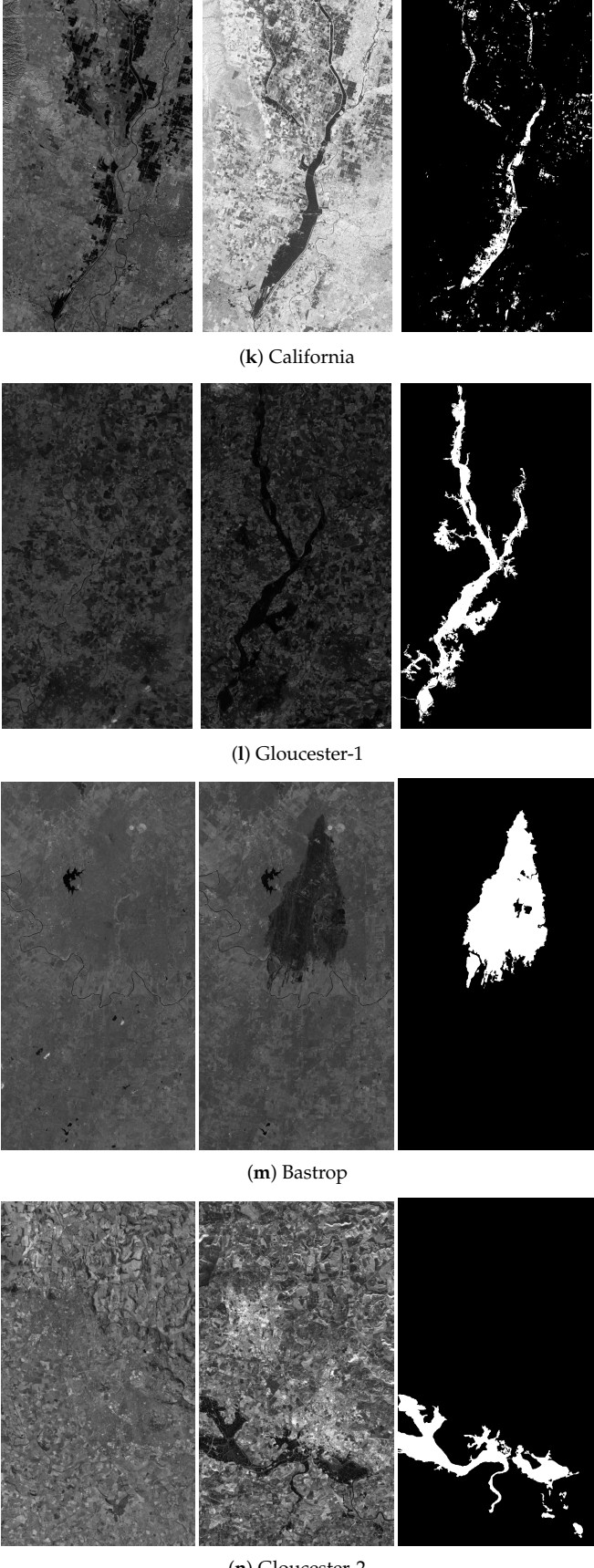

(**k**) California

(**l**) Gloucester-1

(**m**) Bastrop

(**n**) Gloucester-2

**Figure 4.** Datasets used to test the proposed methodology for change detection. From left to right: pre-event, post-event, and reference change map images.

The authors of [38] provided a dataset that contains 321, 96, and 72 images, as well as biomass measurements for vegetation, reproductive, and ripening stages, respectively (see Figure 5). The biomass (ground truth) associated with each image is defined as follows: For each plot of the crop, one linear meter of the plant was cut from the ground. Plants were sampled and weighed (fresh weight), then put in an oven at 65 °C for four days, or until a constant weight was reached. Later, the vegetative biomass (leaves and stems) was separated from the reproductive biomass (panicles and grains). Both vegetative and reproductive biomass were then weighed (dry weight). The images were taken by a UAV equipped with the Tetracam ADC-lite multispectral camera capable of capturing images up to 72.26 mm/pixel in resolution flying at an altitude of 122 m. In our study, the UAV took images of the rice crops, flying over them at a steady altitude of 12 m above ground level (5.93 mm/pixel of resolution). The images (resolution of 1280 × 960) were co-registered, and the bands used to extract the information from the crops were NIR, red, and green.

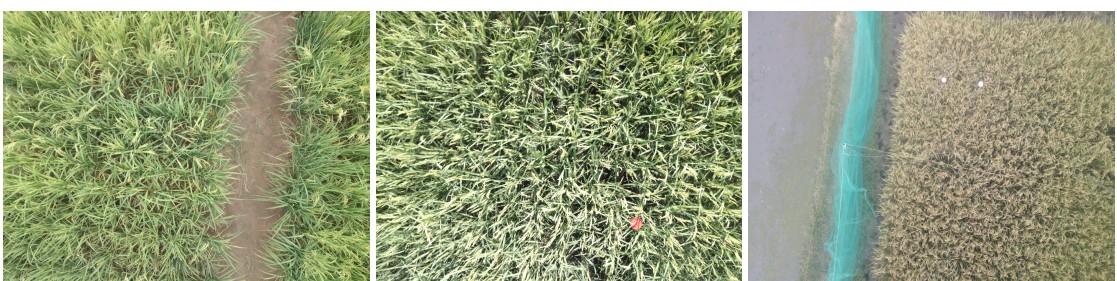

**Figure 5.** Images from left to right represent the stages of the crop: vegetative, reproductive, and ripening, respectively for the genotype Tropical Japonica sub-species.

*2.5. Experimental Setup*

We ran all the codes (to ensure the reproducibility of the proposed method, the code and all datasets are publicly available at: https://github.com/DavidJimenezS/GBF-CD) using a server with two processors, Intel(R) Xeon(R) CPU E5-2650 v4 @ 2.20GHz, with a total of 24 physical cores, 48 threads of processes, and 252 GB of memory RAM @2400 MHz.

2.5.1. Change Detection

We compared the proposed graph-based fusion (GBF)-CD with the classical Kittler–Illingworth (KI) [16] and state-of-the-art techniques: Rayleigh-Rice (rR) [17], Rayleigh-Rayleigh-Rice (rrR) [18], and unsupervised change detection using the regression homogeneous pixel transformation (U-CD-HPT) (code available at https://github.com/llu025/Heterogeneous_CD) [31]. We evaluated each change map generated by all the methods with respect to the ground truth by using relevant metrics in change detection such as: false negatives (FNs), false positives (FPs), precision (P, Equation (8)), recall (R, Equation (9)), Cohen's kappa ($\kappa$, Equation (6)), and overall error (OE), where the metrics FN, FP, and OE are measured in percentage with respect to the number of real change pixels, real non-change pixels, and all the pixels in the image, respectively. The method U-CD-HPT is the only one that requires a post-processing stage by filtering and thresholding to build the change map. We selected the parameters for this post-processing stage according to the values presented by the authors in [31].

The metrics are expressed as follows:

$$\kappa = \frac{p_o - p_e}{1 - p_e},$$

(6)

where $p_o$ is the observed agreement between predictions and labels (the overall accuracy), while $p_e$ is the probability of random agreement, which is estimated from the observed true positives (*TPs*), true negatives (*TNs*), false positives (*FPs*), and false negatives (*FNs*) as:

$$p_e = \left( \frac{TP + FP}{N} \frac{FN + TN}{N} \right) + \left( \frac{TP + FN}{N} \frac{FP + TN}{N} \right). \tag{7}$$

Precision and recall measure the agreement between the predicted and the real changed pixels as:

$$P = \frac{TP}{TP+FP}. \tag{8}$$
$$R = \frac{TP}{TP+FN}. \tag{9}$$

The number of samples ($n_s$) taken by spatial grid sampling and the standard deviations ($\sigma$) for the kernels were set through exhaustive grid-search using *MATLAB*®2017*a*. Table 3 shows the parameter values of the proposed method for each database:

**Table 3.** Parameters used for evaluation of the datasets. The superscripts 1 and 2 stand for pre- and post-event, respectively.

| Database | $n_s$ | $\sigma^1$ | $\sigma^2$ |
|---|---|---|---|
| Mulargia | 93 | $2.5299 \times 10^{-10}$ | $1.5561 \times 10^{-10}$ |
| Omodeo | 93 | $2.7930 \times 10^{-11}$ | $1.6533 \times 10^{-10}$ |
| Alaska | 2 | $1.3720 \times 10^{-9}$ | $-6.7521 \times 10^{-10}$ |
| Madeirinha | 9 | $1.3841 \times 10^{-5}$ | $7.5380 \times 10^{-9}$ |
| Katios National Park | 60 | $1.0319 \times 10^{-13}$ | $-3.2947 \times 10^{-15}$ |
| Atlantico | 240 | $0.0012$ | $-2.6971 \times 10^{-6}$ |
| San Francisco [3] | 4 | $8.3849 \times 10^{-9}$ | $7.5754 \times 10^{-7}$ |
| WenChuan | 39 | $-5.6319 \times 10^{-8}$ | $7.6359 \times 10^{-7}$ |
| Toulouse | 96 | $-8.9790 \times 10^{-15}$ | $-1.4351 \times 10^{-14}$ |
| Prince George | 110 | $-1.9516 \times 10^{-12}$ | $2.6925 \times 10^{-9}$ |
| California [4] | 270 | $-4.7062 \times 10^{-14}$ | $1.9471 \times 10^{-16}$ |
| Gloucester-1 | 12 | $-3.5108 \times 10^{-11}$ | $-1.0611 \times 10^{-10}$ |
| Bastrop | 96 | $-1.2140 \times 10^{-9}$ | $-3.6741 \times 10^{-11}$ |
| Gloucester-2 | 76 | $-7.7131 \times 10^{-13}$ | $-1.6947 \times 10^{-14}$ |

[3] Available at http://earth.esa.int/ers/ers_action/SanFrancisco_SAR_IM_Orbit_47426_20040516.html;
[4] Available at https://sites.google.com/view/luppino/data.

### 2.5.2. Estimating Biomass in Rice Crops

The number of samples ($n_s$) was set to 100, and they were selected using a grid mesh on the image. We used cosine distance for *t*-SNE. For both *t*-SNE and PCA, the dimension *z* and the standard deviations ($\sigma$) for the kernels were set through exhaustive grid-search using MATLAB®2017*a*, which gave us the dimensions $z = 16$. Table 4 shows the mean results of $\sigma$ parameters for each stage of the crop.

**Table 4.** Parameters used to evaluate the datasets. The superscripts 1, 2, and 3 stand for bands R, G, and NIR, respectively.

| Stage | $\bar{\sigma}^1$ | $\bar{\sigma}^2$ | $\bar{\sigma}^3$ |
|---|---|---|---|
| Vegetative | $1.0490 \times 10^{-14}$ | $0.9850 \times 10^{-14}$ | $1.2650 \times 10^{-14}$ |
| Reproductive | $1.0290 \times 10^{-14}$ | $0.7080 \times 10^{-14}$ | $1.1260 \times 10^{-14}$ |
| Ripening | $1.1090 \times 10^{-14}$ | $0.7840 \times 10^{-14}$ | $1.4260 \times 10^{-14}$ |

In order to evaluate the performance of the proposed features for biomass estimation, we used cross-validation splitting the data into training (70%) and testing (30%) datasets. The model considers the whole growth stage of rice crops (vegetative, reproductive, and ripening). To measure the accuracy of the proposed features and the commonly used vegetation indices for biomass estimation, we calculated the root mean squared error (Equation (10)):

$$RMSE = \sqrt{\frac{1}{n}\sum_{i=1}^{m}(y_i - \hat{y}_i)^2},$$

(10)

where $y_i$ are for the real values of the biomass and $\hat{y}_i$ are the estimations of the model.

## 3. Results and Discussion

### 3.1. Change Detection

The visual comparison of the estimated change maps and the corresponding ground truths provide a qualitative assessment of the performance for each method.

Figure 6 illustrates the resulting change maps for the same geographical area, in which each row represents a dataset and each column is one of these methods: KI [16], rR-EM [17], rrR-EM [18], U-CD-HPT [31], and the proposed GBF-CD, respectively. The change maps that were obtained for all the methods show that the most challenging datasets were the Katios National Park and Atlantico (see the fifth and sixth row in Figure 6). The images corresponding to pre- and post-events have similar variability in their pixel intensities. Therefore, the assumption of the probabilistic approaches [17,18] (that the data follow a certain distribution for non-change and change pixels) does not hold. For both the thresholding algorithm (KI) [16] and the unsupervised method based on image-to-image translation (U-CD-HPT) [31], the estimated thresholding and the Frobenius distance between affinity matrices were unable to detect real change. This is because of the similarity between the distributions of change and no-change pixel intensities. In contrast, in the proposed GBF-CD method, the results came from building a fused graph (that minimized the similarities between the pixel intensities in the pre-event and post-event images) and from selecting an approximated eigenvector. This methodology maximizes the mutual information with a prior change map and yields change maps with lower false negative rates.

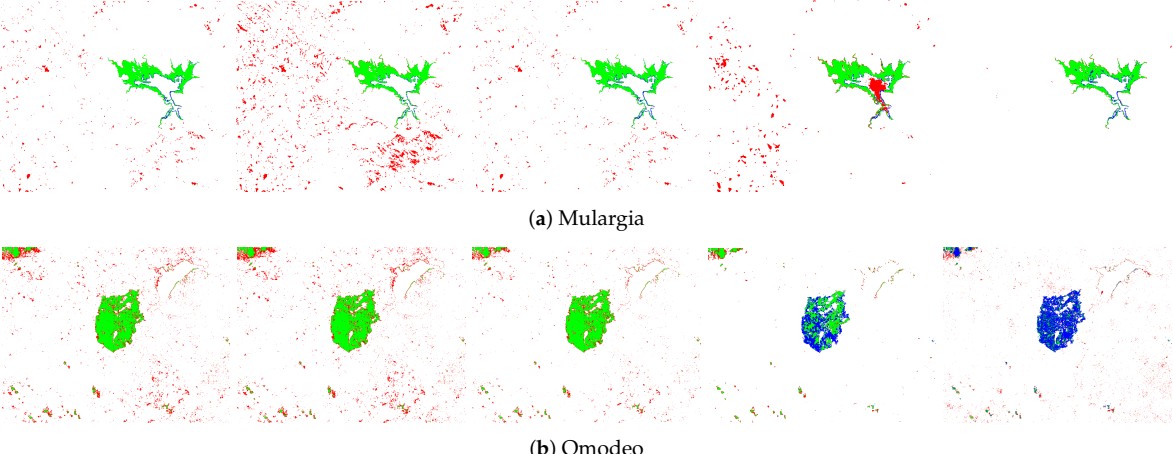

(**a**) Mulargia

(**b**) Omodeo

**Figure 6.** *Cont.*

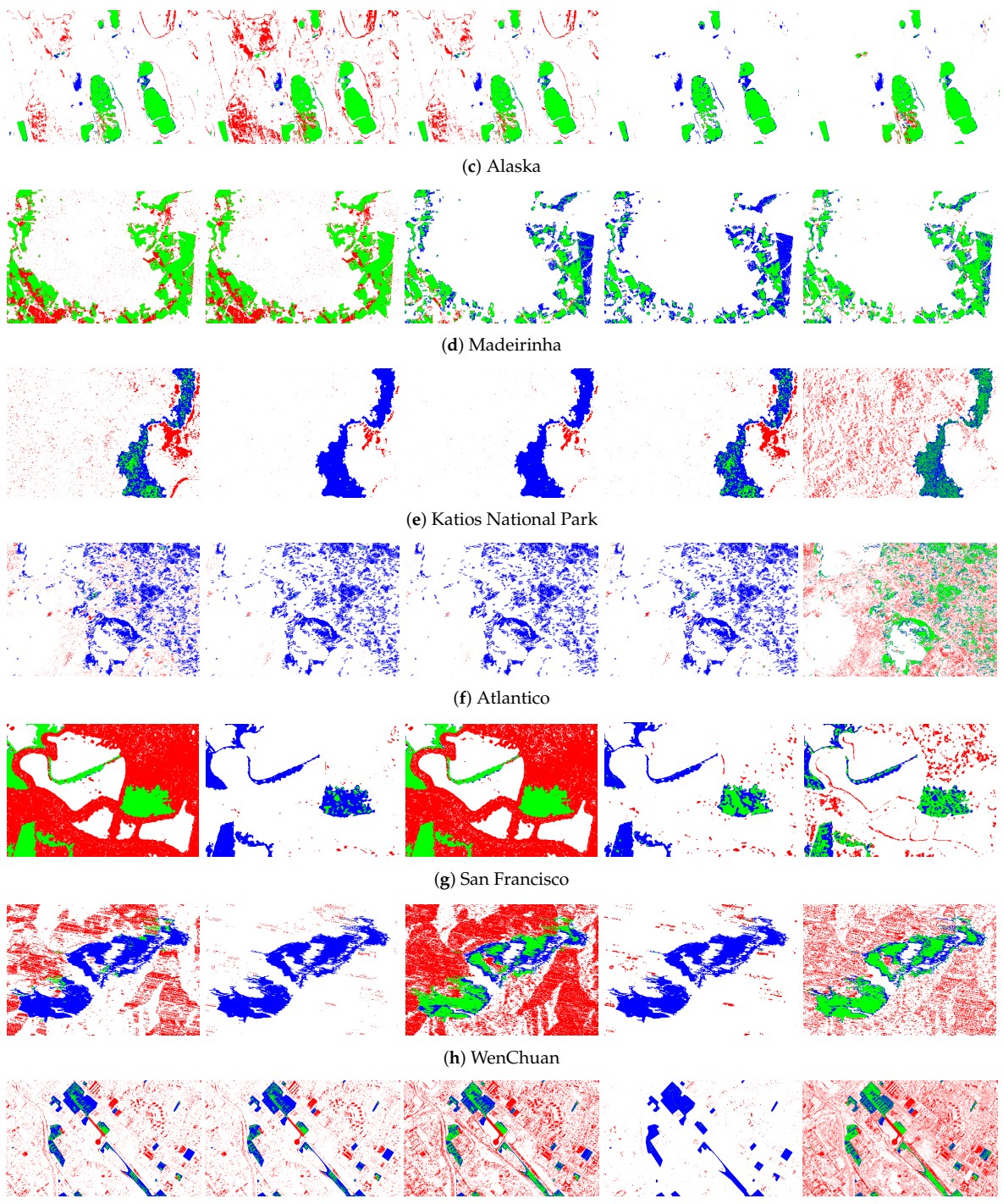

(**c**) Alaska

(**d**) Madeirinha

(**e**) Katios National Park

(**f**) Atlantico

(**g**) San Francisco

(**h**) WenChuan

(**i**) Toulouse

**Figure 6.** *Cont.*

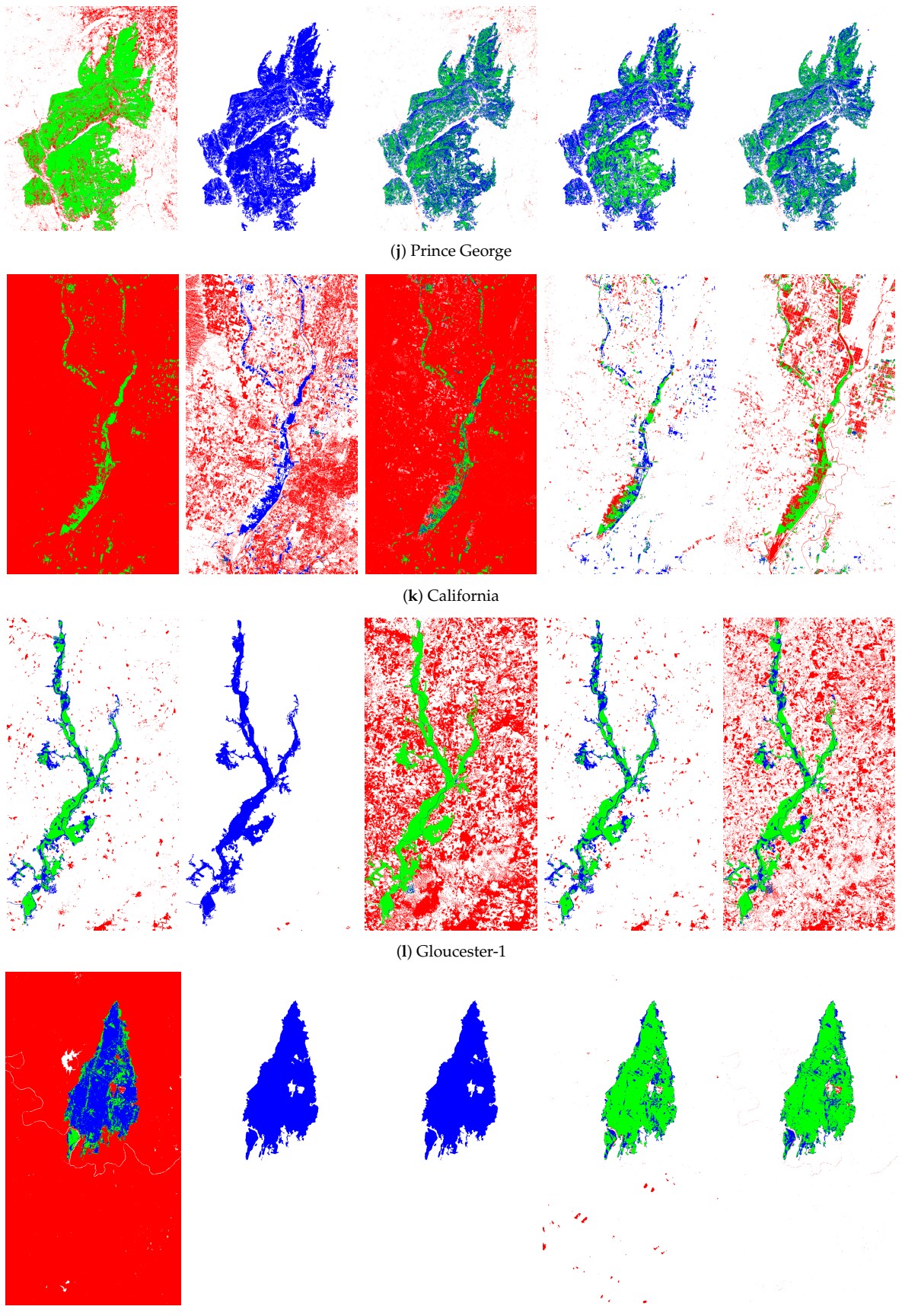

(**j**) Prince George

(**k**) California

(**l**) Gloucester-1

(**m**) Bastrop

**Figure 6.** *Cont.*

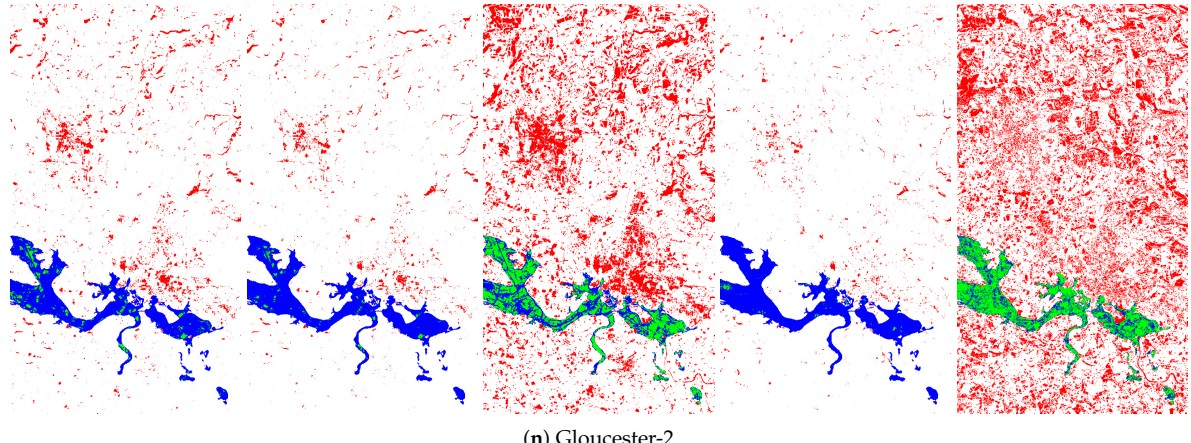

(**n**) Gloucester-2

**Figure 6.** Change detection maps highlighting the false negatives (FNs), false positives (FPs), and correct changed pixels (Cs). Each row corresponds to a dataset and each column to a method: Kittler–Illingworth (KI), Rayleigh-Rice (rR)-EM, Rayleigh-Rayleigh-Rice (rrR)-EM, unsupervised change detection using the regression homogeneous pixel transformation (U-CD-HPT), and graph based fusion (GBF)-CD.

In terms of false negatives (FNs) and false positives (FPs), the probabilistic method (rR) [17] provided the worst performance. This was because the assumption of a large difference between pre-event and post-event images was not true in some of the test scenarios. The KI [16] and rrR-EM [18] algorithms classified all pixels in the San Francisco and California scenarios as belonging to the change category, producing zero FN and very high FP rates. In summary, the U-CD-HPT [31] and GBF-CD methods provide a reasonable compromise between the correctly detected change pixels, FNs, and FP rates (see Tables 5–18, where the best performance with respect to the metrics is written in bold type).

**Table 5.** Performance of the models for the Mulargia dataset. OE, overall error.

| Model | FN (%) | FP (%) | Recall (%) | Precision (%) | $\kappa$ (%) | OE (%) | Time (s) |
|---|---|---|---|---|---|---|---|
| KI [16] | 10.24 | 1.04 | 72.30 | 89.76 | 79.41 | 1.32 | 1.467 |
| rR-EM [17] | **5.72** | 4.01 | 41.73 | **94.28** | 56.05 | 4.06 | 9.881 |
| rrR-EM [18] | 10.14 | 1.06 | 72.04 | 89.86 | 79.29 | 1.33 | 13.895 |
| U-CD-HPT [31] | 9.03 | 2.00 | 58.12 | 90.96 | 69.84 | 2.20 | 107.978 |
| **GBF-CD** | 12.33 | **0.17** | **93.96** | 87.67 | **90.43** | **0.53** | 19.208 |

**Table 6.** Performance of the models for the Omodeo dataset.

| Model | FN (%) | FP (%) | Recall (%) | Precision (%) | $\kappa$ (%) | OE (%) | Time (s) |
|---|---|---|---|---|---|---|---|
| KI [16] | **0.00** | 3.42 | 59.04 | **1.00** | 72.62 | 3.26 | 4.850 |
| rR-EM [17] | 0.01 | 3.73 | 56.93 | **1.00** | 70.80 | 3.56 | 14.489 |
| rrR-EM [18] | 0.01 | 2.14 | 69.73 | **1.00** | **81.12** | **2.04** | 9.928 |
| U-CD-HPT [31] | 45.88 | **0.55** | **82.90** | 54.11 | 64.14 | 2.68 | 294.320 |
| **GBF-CD** | 77.00 | 1.26 | 47.26 | 22.99 | 28.73 | 4.83 | 91.624 |

**Table 7.** Performance of the models for the Alaska dataset.

| Model | FN (%) | FP (%) | Recall (%) | Precision (%) | κ (%) | OE (%) | Time (s) |
|---|---|---|---|---|---|---|---|
| KI [16] | 14.13 | 3.57 | 74.23 | 85.86 | 76.98 | 4.70 | 1.424 |
| rR-EM [17] | **8.07** | 10.91 | 50.24 | **91.92** | 59.34 | 10.60 | 7.638 |
| rrR-EM [18] | 12.52 | 4.81 | 68.51 | 87.48 | 73.68 | 5.64 | 8.322 |
| U-CD-HPT [31] | 22.01 | **0.15** | **98.38** | 77.98 | 85.65 | 2.49 | 123.214 |
| **GBF-CD** | 11.66 | 0.87 | 92.36 | 88.34 | **89.17** | **2.02** | 3.623 |

**Table 8.** Performance of the models for the Madeirinha dataset.

| Model | FN (%) | FP (%) | Recall (%) | Precision (%) | κ (%) | OE (%) | Time (s) |
|---|---|---|---|---|---|---|---|
| KI [16] | **0.01** | 10.44 | 69.47 | **99.99** | 76.70 | 8.44 | 1.347 |
| rR-EM [17] | **0.01** | 10.19 | 69.98 | **99.99** | 77.18 | 8.23 | 6.171 |
| rrR-EM [18] | 40.31 | 1.32 | 91.45 | 59.69 | 67.27 | 8.81 | 16.320 |
| U-CD-HPT [31] | 61.05 | **0.11** | **98.78** | 38.94 | 50.48 | 11.81 | 77.366 |
| **GBF-CD** | 24.44 | 1.13 | 94.06 | 75.56 | **80.46** | **5.60** | 4.100 |

**Table 9.** Performance of the models for the Katios dataset.

| Model | FN (%) | FP (%) | Recall (%) | Precision (%) | κ (%) | OE (%) | Time (s) |
|---|---|---|---|---|---|---|---|
| KI [16] | 67.88 | 5.87 | 39.20 | 32.12 | 28.51 | 12.42 | 1.769 |
| rR-EM [17] | 99.84 | **1.18** | 1.49 | 0.15 | -1.72 | 11.60 | 4.013 |
| rrR-EM [18] | 99.79 | 1.29 | 1.85 | 0.21 | -1.79 | 11.69 | 4.083 |
| U-CD-HPT [31] | 73.00 | 3.58 | **47.03** | 26.99 | 28.82 | **10.90** | 457.230 |
| **GBF-CD** | **52.05** | 10.63 | 34.74 | **47.95** | **31.96** | 15.00 | 34.481 |

**Table 10.** Performance of the models for the Atlantico dataset.

| Model | FN (%) | FP (%) | Recall (%) | Precision (%) | κ (%) | OE (%) | Time (s) |
|---|---|---|---|---|---|---|---|
| KI [16] | 98.34 | 3.00 | 9.12 | 1.65 | -2.03 | 17.72 | 1.652 |
| rR-EM [17] | 99.69 | 0.29 | 15.70 | 0.30 | 0.01 | 15.63 | 5.099 |
| rrR-EM [18] | 99.93 | **0.08** | 11.62 | 0.06 | -0.04 | **15.49** | – |
| U-CD-HPT [31] | 99.13 | 0.28 | 36.01 | 0.86 | 0.97 | 15.53 | 333.742 |
| **GBF-CD** | **30.42** | 13.69 | **48.11** | **69.57** | **47.26** | 16.27 | 103.911 |

**Table 11.** Performance of the models for the San Francisco dataset.

| Model | FN (%) | FP (%) | Recall (%) | Precision (%) | κ (%) | OE (%) | Time (s) |
|---|---|---|---|---|---|---|---|
| KI [16] | **1.08** | 63.16 | 18.55 | **98.92** | 12.54 | 55.28 | 1.315 |
| rR-EM [17] | 92.75 | **0.59** | 64.05 | 7.24 | 10.71 | 12.29 | 3.282 |
| rrR-EM [18] | 2.19 | 61.23 | 18.85 | 97.80 | 13.11 | 53.73 | 3.813 |
| U-CD-HPT [31] | 75.81 | 1.52 | **69.62** | 24.19 | 31.43 | **10.92** | 64.899 |
| **GBF-CD** | 48.82 | 7.64 | 49.34 | 51.17 | **42.85** | 12.87 | 3.213 |

**Table 12.** Performance of the models for the WenChuan dataset.

| Model | FN (%) | FP (%) | Recall (%) | Precision (%) | κ (%) | OE (%) | Time (s) |
|---|---|---|---|---|---|---|---|
| KI [16] | 93.29 | 22.11 | 5.94 | 6.70 | -14.67 | 34.38 | 1.380 |
| rR-EM [17] | 99.79 | **1.07** | 3.72 | 0.20 | -1.41 | **18.10** | 3.318 |
| rrR-EM [18] | 41.61 | 53.95 | 18.40 | 58.39 | 2.38 | 51.83 | 3.678 |
| U-CD-HPT [31] | 99.69 | 2.06 | 3.00 | 0.30 | -2.73 | 18.88 | 65.025 |
| **GBF-CD** | **35.82** | 22.52 | **37.25** | **64.17** | **32.39** | 24.81 | 6.235 |

**Table 13.** Performance of the models for the Toulouse dataset.

| Model | FN (%) | FP (%) | Recall (%) | Precision (%) | κ (%) | OE (%) | Time (s) |
|---|---|---|---|---|---|---|---|
| KI [16] | 74.42 | 8.33 | 20.97 | 25.57 | 15.66 | 13.59 | 1.380 |
| rR-EM [17] | 74.94 | 8.11 | **21.07** | 25.05 | 15.59 | 13.43 | 3.318 |
| rrR-EM [18] | **52.32** | 22.07 | 15.74 | **47.67** | 13.29 | 24.47 | 3.678 |
| U-CD-HPT [31] | 98.30 | **0.97** | 13.11 | 1.69 | 1.20 | **8.72** | 4449.601 |
| **GBF-CD** | 54.27 | 17.33 | 18.57 | 45.72 | **17.02** | 20.27 | 839.940 |

**Table 14.** Performance of the models for the Prince George dataset.

| Model | FN (%) | FP (%) | Recall (%) | Precision (%) | κ (%) | OE (%) | Time (s) |
|---|---|---|---|---|---|---|---|
| KI [16] | **0.60** | 16.15 | 70.23 | **99.39** | **73.79** | **11.84** | 2.575 |
| rR-EM [17] | 100.00 | **0.00** | – | 0.00 | 0.00 | 27.71 | 4.764 |
| rrR-EM [18] | 54.01 | 1.13 | 93.93 | 45.99 | 53.22 | 15.79 | 723.778 |
| U-CD-HPT [31] | 61.23 | 0.20 | **98.61** | 38.76 | 47.42 | 17.12 | 2075.130 |
| **GBF-CD** | 54.10 | 0.38 | 97.86 | 45.90 | 54.42 | 15.27 | 240.742 |

**Table 15.** Performance of the models for the California dataset.

| Model | FN (%) | FP (%) | Recall (%) | Precision (%) | κ (%) | OE (%) | Time (s) |
|---|---|---|---|---|---|---|---|
| KI [16] | **0.17** | 99.97 | 4.36 | **99.83** | −0.01 | 95.61 | 2.910 |
| rR-EM [17] | 97.74 | 31.71 | 0.32 | 2.26 | −7.66 | 34.60 | 9.989 |
| rrR-EM [18] | 18.01 | 97.85 | 3.69 | 81.98 | −1.43 | 94.36 | 25.521 |
| U-CD-HPT [31] | 58.21 | **2.79** | **40.59** | 41.79 | **38.45** | **5.21** | 2955.937 |
| **GBF-CD** | 11.93 | 11.79 | 25.44 | 88.06 | 35.07 | 11.80 | 921.624 |

**Table 16.** Performance of the models for the Gloucester-1 dataset.

| Model | FN (%) | FP (%) | Recall (%) | Precision (%) | κ (%) | OE (%) | Time (s) |
|---|---|---|---|---|---|---|---|
| KI [16] | 43.16 | 2.33 | **69.62** | 56.83 | **59.44** | **5.85** | 2.933 |
| rR-EM [17] | 99.99 | **0.04** | 0.03 | 0.00 | -0.07 | 8.64 | 8.202 |
| rrR-EM [18] | **2.35** | 44.06 | 17.27 | **97.65** | 17.24 | 40.47 | 24.540 |
| U-CD-HPT [31] | 44.60 | 2.41 | 68.31 | 55.39 | 57.94 | 6.05 | 3808.564 |
| **GBF-CD** | 23.80 | 26.57 | 21.26 | 76.19 | 22.86 | 26.33 | 96.464 |

**Table 17.** Performance of the models for the Bastrop dataset.

| Model | FN (%) | FP (%) | Recall (%) | Precision (%) | κ (%) | OE (%) | Time (s) |
|---|---|---|---|---|---|---|---|
| KI [16] | 73.30 | 99.16 | 3.10 | 26.69 | -16.67 | 96.41 | 1.380 |
| rR-EM [17] | 100.00 | **0.00** | – | 0.00 | 0.00 | 10.63 | 3.318 |
| rrR-EM [18] | 100.00 | **0.00** | – | 0.00 | 0.00 | 10.63 | 3.678 |
| U-CD-HPT [31] | **15.50** | 0.39 | 96.17 | **84.49** | **88.84** | 2.00 | 365.296 |
| **GBF-CD** | 16.83 | 0.23 | **97.71** | 83.16 | 88.75 | **1.99** | 109.347 |

**Table 18.** Performance of the models for the Gloucester-2 dataset.

| Model | FN (%) | FP (%) | Recall (%) | Precision (%) | κ (%) | OE (%) | Time (s) |
|---|---|---|---|---|---|---|---|
| KI [16] | 90.34 | 4.25 | 13.46 | 9.65 | 6.21 | 9.78 | 1.380 |
| rR-EM [17] | 96.29 | 2.33 | 9.80 | 3.70 | 1.92 | 8.36 | 3.318 |
| rrR-EM [18] | 44.12 | 19.72 | **16.26** | 55.87 | **16.93** | 21.29 | 3.678 |
| U-CD-HPT [31] | 98.36 | **1.57** | 1.63 | 6.63 | 0.08 | **7.78** | 3767.047 |
| **GBF-CD** | **29.39** | 27.71 | 14.86 | **70.60** | 15.62 | 27.82 | 543.650 |

The Toulouse, California, Bastrop, and Gloucester-2 test scenarios are represented by NIR and SAR images. With regard to the Toulouse and Gloucester-2 datasets, the rrR-EM and GBF-CD methods yielded change maps with high TPs, low FNs, and high FP rates. In contrast, KI, rR-EM, and U-CD-HPT provided low TPs and high FN rates. In the case of the California dataset, the KI, rR-EM, and rrR-EM algorithms provided inaccurate change maps due to the fact that these methods were devised for processing homogeneous (one modality) input data. Despite the data heterogeneity, the U-CD-HPT [31] and GBF-CD algorithms provided better performance in terms of FNs, FPs, and $\kappa$. Unlike the KI, rR-EM, and rrR-EM methods, the algorithms U-CD-HPT and GBF-CD used for the Bastrop dataset yielded an accurate change map.

To illustrate the relative performance of each CD method in all the challenging test scenarios, we counted the number of times a given CD method outperformed the competing algorithms in a specific performance metric (see Figure 7). We observed that the proposed GBF-CD method outperformed (in terms of $\kappa$) the competing algorithms in eight (Mulargia, Alaska, Madeirinha, Katios, Atlantico, San Francisco, WenChuan, and Toulouse) of the fourteen datasets. Moreover, the GBF-CD algorithm achieved the best performance metrics (FN, recall, precision, and OE) in four (Katios, Atlantico, WenChuan, and Gloucester-2) of the test scenarios. It also showed the lowest FP rate in one scenario (Mulargia). Overall, the proposed GBF-CD algorithm outperformed the comparison methods in at least one performance metric. In contrast, the KI, rR-EM, rrR-EM, and U-CD-HPT algorithms did not surpass other competing methodologies in at least one performance metric.

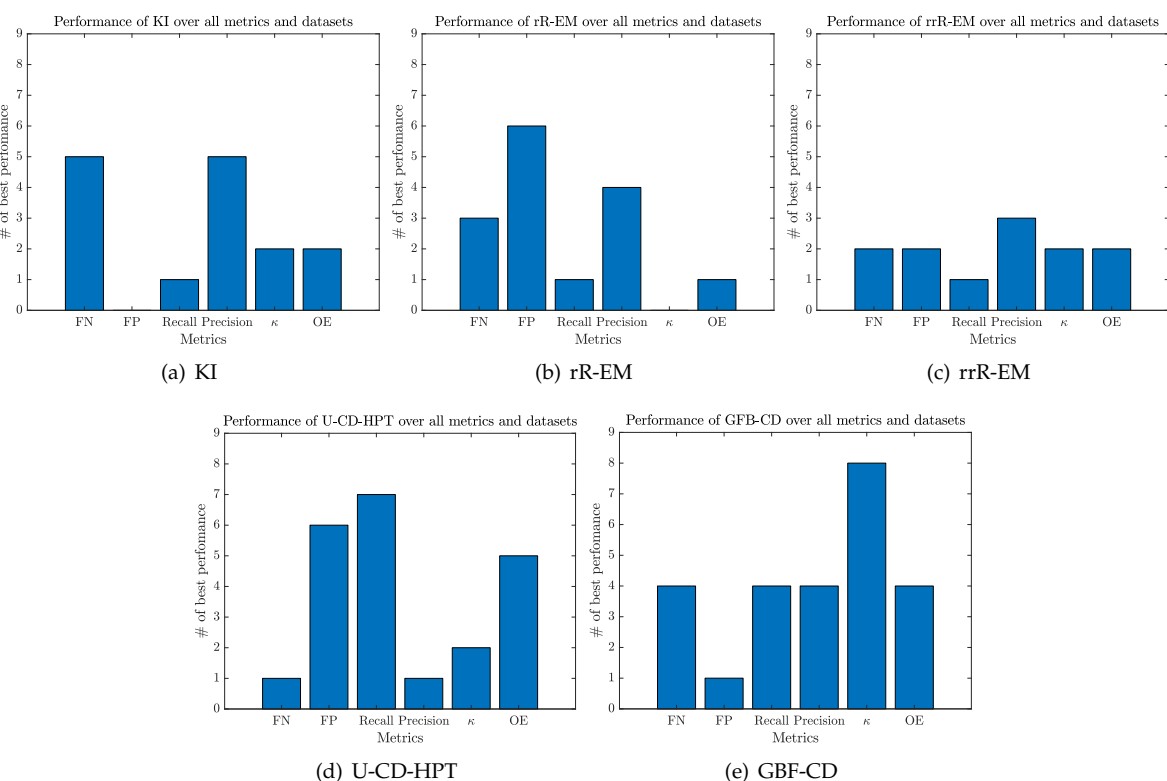

**Figure 7.** Bar charts that evaluate the performance of each method over all metrics and datasets. The count for each method in one of the six possible metrics means that in one dataset, the model outperformed all the competing methods in that metric.

### 3.2. Biomass Estimation

Figure 8 illustrates the comparison of the biomass prediction results. This was achieved by applying the dimensionality reduction techniques *t*-SNE and PCA to the features extracted from the proposed graph-based fusion approach, in addition to the biomass estimation yielded by using the

traditional VIs. These results show that the VI does not capture the biomass features during the growth of rice crops. In contrast, the regressor that was trained with the features obtained after applying the dimensionality reduction techniques provided better prediction results and lower estimation errors (as shown in Table 19).

Even though the proposed graph-based fusion features outperformed the traditional VIs for biomass estimation, there is still a need for further work; for instance, to decrease the computation time, as it currently takes approximately three hours to extract the features and train the models. It would also be advantageous to reduce the dependency of the performance on the number of selected samples $n_s$ and the standard deviation $\sigma$ and explore parameter tuning methods beyond exhaustive grid search. However, one regression model based on the proposed features predicted the biomass well, despite its variability during different growth stages of rice crops. This is a result of the fact that the graph-based features capture both radiometric and structurally useful information from the MS bands. In contrast, the VIs are not able to capture the biomass variability for rice crop growth, requiring three separate regression models [38].

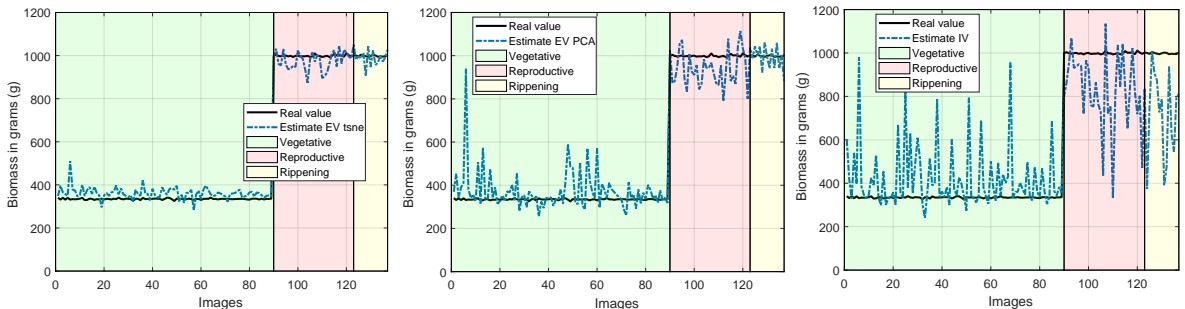

**Figure 8.** Regression performance by one model for all rice crop growth stages. From left to right, the models are: *t*-SNE, PCA, and vegetation indices (VIs).

**Table 19.** Performance of each model for biomass estimation. The evaluated metric is the root mean squared error (RMSE).

|  | VI | PCA | t-SNE |
|---|---|---|---|
| **RMSE** | 213.290 | 95.795 | **40.273** |

## 4. Conclusions

In this paper, we introduced a graph-based data fusion methodology for remote sensing images and tested it in two applications: change detection and biomass estimation in rice crops. The main contribution of this study was a "data-driven" framework used to capture unique information for multi-temporal, multi-spectral, and multi-modal/heterogeneous (Toulouse, California, Bastrop, and Gloucester-2 datasets) images in a fused graph. The fused graph stage captures information in one graph from a small set of samples (less than 10% of the total pixels) for each dataset (in different times or bands for homogeneous or heterogeneous data).

For the change detection application, we utilized a mutual information criterion to select from a prior and an eigen-image to build the final change map. In this case, our method is parametric since it depends on a number of samples and the prior information (difference images). Thereby, from the results for all datasets, we observed that our model obtained coherent change maps and outperformed state-of-the-art methods [16–18]. The method proposed in this study performed well with respect to the metrics TP and FN in multi-sensor datasets such as: Toulouse, California, Bastrop, and Gloucester-2. In addition, the model developed in this paper does not require a post-processing stage, such as that needed by the U-CD-HPT method.

In biomass estimation, the model showed that the features extracted from the fused graph with a dimensionality reduction technique (i.e., PCA or *t*-SNE) capture the variability of biomass in rice crops. This makes it possible to predict the biomass features throughout the growth stages in rice crops, by using one regression model. These outcomes are more comprehensive than those reported by the authors in [38], in which three separate regression models estimated the biomass at each stage of the rice crop, based on VI features.

Future studies are necessary to reduce the dependency of the proposed method on the manual selection of $n_s$ samples and prior information, currently defined in terms of the differences between the pre-event and post-event images.

**Author Contributions:** D.A.J.-S. proposed the original idea, designed the studies, performed the experiments, and analyzed the data. H.D.B.-R., H.D.V.-C., and J.C. contributed significantly to the discussion of the results. D.A.J.-S. wrote the manuscript, which was revised by all authors. All authors read and agreed to the published version of the manuscript.

**Funding:** This work was funded by the OMICAS program: *Optimización Multiescala In-silico de Cultivos Agrícolas Sostenibles (Infraestructura y validación en Arroz y Caña de Azúcar)*, anchored at the Pontificia Universidad Javeriana in Cali and funded within the Colombian Scientific Ecosystem by The World Bank, the Colombian Ministry of Science, Technology and Innovation, the Colombian Ministry of Education, the Colombian Ministry of Industry and Tourism, and ICETEX under grant ID: FP44842-217-2018.

**Acknowledgments:** The authors would like to thank the professors Julian Colorado and Ivan Mondragon for their support with the image dataset collection and all CIAT staff that supported the experiments over the crops located at CIAT headquarters in Palmira, Valle del Cauca, Colombia; in particular, Yolima Ospina and Cecile Grenier for their support in upland and lowland trials and Luigi Tommaso Luppino [31] for sharing the code and supporting us in the replication of his results.

**Conflicts of Interest:** The authors declare no conflict of interest.

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
