# Peer review of "Graph-Based Data Fusion Applied to: Change Detection and Biomass Estimation in Rice Crops"

_remotesensing, doi:10.3390/rs12172683_

Round 1

Reviewer 1 Report

The English grammar must be improved, being specially careful with verbal tenses, number concordances, etc., there are so many mistakes that is surprising that the document was reviewed by all the authors before submission (as the Author Contributions states). Just some examples are:
- First sentence in the introduction is not grammatically correct.
- Grammatical error: "For instances, in [9] was proposed"
- "The authors segments urban areas through K-means on the eigenvectors of the fused graph and then classifying the areas"
- Right before Table 1, where it says "The data sets includes events" it should say "The data sets include events"
- Line 196, "We compare" should be "We compared"

And many others...

There are some things that make this paper difficult to read. The concept of data fusion used is confusing. Reading the title and abstract one thinks that different sensor images were going to be fused (which os the main standard remote sensing data fusion concept), but at the end, as far as I understand, there is not fusion at all, but features extracted using graphs concept.
- What's the meaning of "each sensor represents some new degrees of freedom"? and "modalities", is it the same as sensor?

The Introduction section lacks an analysis of the state of the art of methodologies used in a similar way as in this paper, applied to change detection and biomass estimation.
Some parts are out of place in this section, e.g.:
- Lines 70-72 and 88-90 correspond to Methods, more than to Introduction.
- Lines 73-76 and 91-92 belong to Results, more thatn to Introduction.They could be in the abstract, but not in this section.

The dataset used for evaluation is large but the type of changes chosen are very obvious and "easy" to detect.
Reference data used for evaluation of crop biomass is not sufficiently explained: In section 2.2, authors say "The measurement of biomass on rice crops have relied on destructive sampling or remote sensing images" (remote sensing images?). I imagine that this is not your reference data, but something general that can be placed in the introduction secion... What is exactly the accumulated plant biomass? It is poorly explained in lines 151 to 156, but it should be actually described in a section of Data.
Then, authors say "The authors of [35] provided the data set that contains 321, 96, and 72 images and biomass 189 measurements for vegetation, reproductive, and ripening stages, respectively" Are those the reference evaluation data used? This is hard to understand and the methodology section needs to be described in a better organized manner. I propose the authors to first clearly describe the goals, then the data used (i) to apply the methods and (ii) for evaluation, and then to describe the methods applied
On the other hand, what kind of biomass has been predicted, stage 1, 2 or 3?
The only comment done about sampling is "The number of samples (ns) was set to 100", but how were the samples? how were they taken? This needs to be fully described.
At the end, it is hard to know how the experiments were evaluated.

What are the units for overall error, percentage? It is surprisingly variable.

Since goals and data are so poorly described then methods are difficult to follow and results are not appealing to me. I suggest a reorganization of the first part of the paper so it can be properly followed.

Author Response

Thank you for your reviews and very valuable comments and suggestions, this will indeed highlight the contribution of our work. It helped us to prepare a new improved version of the draft. The detailed answers are listed in the pdf attached.

Reviewer 2 Report

The paper looks very interesting, and the applied methods are statistically powerful enough to achieve the objective of the manuscript, however, the structure in which it is presented makes it difficult to read. I think there is a lot of method in the introduction, a lot of results in the methods section, etc.
 Page 7: Sentinel image from 2010-2011 ?, Please clarify if Sentinel 1 A and B were launched 2014 and 2016 respectively.
Page 7: Landsat 5? TM or MSS ??, Landsat TM, Landsat 5 ??. Why some scenes dont have coordinates?

1) L133. Algorithm 1: it is more appropriate and less confusing to talk about standardization / scaling, not normalization (minMax scaling)

2) L.118: What type of sampling did you use to choose the samples? Simple random? (unless Nystrom extension as they say, explain that and infer from there ...)
3) Figure 3 ... is it cited? Divide it into 3 figures? Summarize it? 4) L.158 NDVI is the reason between NIR and RED, please correct. Why didn't  use EVI?
5) L.186 algorithm 2, does not talk about standardization before PCA, but does mention it in the figure (they use zscore). It would be good to explain briefly how it works (as with algorithm 1) and comment on why they used this scaling here and not the same one above (this is not for them, for you: the answer is that in the images the range is bounded 0 to 255, 0 to 2 ^ n, but in the data they are using [filtered], the max / min is not defined.5)
6) L.215 why 16 dimensions? Why not more / less?
Clarify: the objective of the work is not well understood; It is one thing to classify the image and detect the change in use, and the other is to calculate production (biomass).

Author Response

(The authors gave the same response as above.)

Round 2

Reviewer 1 Report

The paper has improved after revision and now is easier to understand. I still found some minor issues to correct:

- Many grammar mistakes yet, some examples are: An SAR, an MS, relating to data, makes it possible to quantify, satellites which provides, They have made it possible to calculate vegetation indices, set out out, an UAV, we calculate(d), and a prior. So I recommend the authors to check the text more carefully by an English-language native.
- MS is not defined the 1st time
- resolution of 1280 x 960) should be size...
- Evaluation: I get confused with MA vs FN, and FA vs FP, are they the same? if so, please avoid use all of them. Also, the MA and FA expressed in % on tables, what represents the total percentage? the total samples? This is not described in the text.

Author Response

Dear reviewer,   Thank you for your reviews and very valuable comments and suggestions. The detailed answers are listed below point by point.

The paper has improved after revision and now is easier to understand. I still found some minor issues to correct:

- Many grammar mistakes yet, some examples are: An SAR, an MS, relating to data, makes it possible to quantify, satellites which provides, They have made it possible to calculate vegetation indices, set out out, an UAV, we calculate(d), and a prior. So I recommend the authors to check the text more carefully by an English-language native.

REPLY: Thank you for your careful reading. We carried out all the suggested corrections and the paper was revised by a professional English-speaking proofreading service.  

- MS is not defined the 1st time

REPLY: Thank you for your comment. We defined it at line 18.

- resolution of 1280 x 960) should be size...

REPLY: We changed it (line 36).

- Evaluation: I get confused with MA vs FN, and FA vs FP, are they the same? if so, please avoid use all of them. Also, the MA and FA expressed in % on tables, what represents the total percentage? the total samples? This is not described in the text.

REPLY: Thank  you for your comment. We replaced FN by MA and FP by FA in equations 7, 8, and 9.  Where the metrics MA, FA, and OE are measured in percentage with respect to the number of real change pixels, real non-change pixels, and all the pixels in the image, respectively (lines 267 to 269).

Reviewer 2 Report

The comments on the manuscript were addressed by the authors, improving the work, which is why I suggest its publication in its current state.

Author Response

Dear reviewer,     Thank you for your reviews and very valuable comments and suggestions. They were important to highlight the contribution of our work.